# Dithiothreitol Affects the Fertilization Response in Immature and Maturing Starfish Oocytes

**DOI:** 10.3390/biom13111659

**Published:** 2023-11-17

**Authors:** Nunzia Limatola, Jong Tai Chun, Kazuyoshi Chiba, Luigia Santella

**Affiliations:** 1Department of Research Infrastructures for Marine Biological Resources, Stazione Zoologica Anton Dohrn, 80121 Napoli, Italy; 2Department of Biology and Evolution of Marine Organisms, Stazione Zoologica Anton Dohrn, 80121 Napoli, Italy; chun@szn.it; 3Department of Biological Sciences, Ochanomizu University, Tokyo 112-8610, Japan; chiba.kazuyoshi@ocha.ac.jp

**Keywords:** starfish, oocyte maturation, disulfide-reducing agents, DTT, actin, calcium, fertilization, polyspermy

## Abstract

Immature starfish oocytes isolated from the ovary are susceptible to polyspermy due to the structural organization of the vitelline layer covering the oocyte plasma membrane, as well as the distribution and biochemical properties of the actin cytoskeleton of the oocyte cortex. After the resumption of the meiotic cycle of the oocyte triggered by the hormone 1-methyladenine, the maturing oocyte reaches fertilizable conditions to be stimulated by only one sperm with a normal Ca^2+^ response and cortical reaction. This cytoplasmic ripening of the oocyte, resulting in normal fertilization and development, is due to the remodeling of the cortical actin cytoskeleton and germinal vesicle breakdown (GVBD). Since disulfide-reducing agents such as dithiothreitol (DTT) are known to induce the maturation and GVBD of oocytes in many species of starfish, we analyzed the pattern of the fertilization response displayed by *Astropecten aranciacus* oocytes pre-exposed to DTT with or without 1-MA stimulation. Short treatment of *A. aranciacus* immature oocytes with DTT reduced the rate of polyspermic fertilization and altered the sperm-induced Ca^2+^ response by changing the morphology of microvilli, cortical granules, and biochemical properties of the cortical F-actin. At variance with 1-MA, the DTT treatment of immature starfish oocytes for 70 min did not induce GVBD. On the other hand, the DTT treatment caused an alteration in microvilli morphology and a drastic depolymerization of the cortical F-actin, which impaired the sperm-induced Ca^2+^ response at fertilization and the subsequent embryonic development.

## 1. Introduction

Cellular and molecular mechanisms regulating the maturation process of oocytes have been highlighted by studies on starfish, whose oocyte maturation is triggered by the maturation-inducing hormone 1-methyladenine (1-MA) produced by ovarian follicle cells. The immature oocytes extracted from the ovary contain a large nucleus (germinal vesicle, GV) at the animal pole and are thus arrested at the prophase of the first meiotic division (GV-stage). The external application of 1-MA induces them to resume the meiotic cycle and undergo oocyte maturation. The hormone acts on an unidentified receptor on the oocyte plasma membrane and restarts the oocyte meiotic cell cycle [1,2]. The definition of the physiological and biochemical steps regulating the signal transduction process has led to the identification of the Cdk1-cyclin B or maturation promoting factor (MPF) [3,4], which induces the germinal vesicle breakdown (GVBD) and the establishment of the optimal fertilizability conditions of the mature eggs. Thus, scientific knowledge has been produced on the changes in the structural organization of the surface and cortex of the polyspermic immature oocytes induced by 1-MA and necessary for normal monospermic fertilization. Specifically, it has been shown that microvilli, cortical granules (CGs), and the cortical actin of the immature oocytes, which are rearranged during the 1-MA triggered maturation process with the contribution of nuclear components following GVBD, are responsible for eliciting normal electrophysiological and Ca^2+^ changes as well as the entry of only one sperm [5,6,7,8,9,10,11,12,13,14,15,16]. 

In starfish, the jelly coat-induced formation of the long acrosomal process of the sperm [11,17,18,19] has allowed following in vivo the electrical and Ca^2+^ increases upon the fusion of the sperm with the egg plasma membrane followed by sperm incorporation [7,11]. Indeed, the tip of the F-actin-based acrosomal process of the fertilizing sperm triggered by components of the jelly coat (JC) can penetrate the vitelline layer (VL) endowed with openings and fuse with the cell plasma membrane [11,13,14,16]. The exceptionality of using starfish oocytes as an animal model is represented by the possibility of fertilizing them before and at later stages of the maturation process [20,21,22,23]. Such studies have added weight to the crucial role played by the structural organization and biochemical properties of cortical actin [24,25,26,27], which are heavily affected in aged and overmatured eggs and, thereby, promote an altered sperm-induced Ca^2+^ signal at fertilization and polyspermic entry [28].

Similar to the action of 1-MA, the external application of disulfide-reducing agents such as dithiothreitol (DTT) has been previously shown to stimulate MPF and meiosis progression in starfish oocytes. Even if the site and mode of the action of DTT may not be the same as those of 1-MA, it was shown that its application induced oocyte maturation acting on the oocyte surface through an increase in the SH content of oocyte cortical proteins following the reduction in the disulfide bonds [29,30,31,32,33]. The acquisition of fertilizability in oocytes treated with DTT was found to be quite similar to that established by 1-MA-induced maturation, as DTT-treated oocytes could be activated upon insemination. At the ultrastructural level, it was shown that DTT had a time-dependent effect on the structural organization of the microvilli and VL, which underwent partial destruction [29]. 

Similarly, in sea urchins, the only effect of 10 min of DTT treatment at a concentration of 10 mM in normal seawater, a procedure used to study the changes in the topography of the sea urchin egg surface following insemination, was to prevent the separation of the VL without affecting the physiology of the fertilization response [34,35,36]. However, a more recent EM analysis combined with live cell imaging experiments showed that DTT treatment at a pH of 7.57 or 9 (the sensitivity of the oocytes to DTT depends on the external pH) heavily affected the topography of the egg surface by inducing a partial destruction of the VL (at pH 7.57) or its complete removal (at pH 9) [37,38]. Furthermore, removing the VL from the surface of unfertilized sea urchin eggs induced polyspermic fertilization. At variance with the prevailing view, results have shown that the plasma membrane, not the VL, is the site for specific recognition between gametes membranes [38,39,40,41,42]. Thus, the structural integrity of the VL covering the unfertilized sea urchin eggs is essential to mask the fusogenic sites residing in the cell plasma membrane, the exposure of which would induce homologous polyspermic fertilization. Furthermore, the assessment of the effect of DTT on the physiological responses of sea urchin eggs to the fertilizing sperm has been shown to be linked to the DTT-induced alteration in the microvillar morphology and the cytoskeletal properties of the egg cortex. The structural modification of the egg ectoplasm, in turn, compromised the Ca^2+^ response and the subsequent polymerization of the cortical actin following the intracellular pH increase [37,43,44,45,46,47,48,49,50].

In the present work, we tested the effect of DTT (10 mM in normal seawater, pH 7.57) on the fertilization response of immature starfish oocytes known to be polyspermic. In addition to reducing the number of sperm incorporation, 10 min DTT treatment modified the Ca^2+^ signals triggered by multiple sperm due to altering the morpho-functionality of the actin of the cortex of the immature oocytes. Upon insemination of GV-stage oocytes pretreated with DTT, the Ca^2+^ response and the rate of sperm incorporation were affected due to the reduction in disulfide bonds of the cytoskeletal proteins of the oocyte cortex. As previously shown, the results of the DTT-mediated oocyte maturation highlighted the biochemical differences between the reduction in the disulfide bonds of cytoskeletal proteins of the oocyte cortex and the natural maturation induced by the hormone 1-MA. Such differences were evidenced in the modality by which the DTT and 1-MA matured oocytes responded to the physiological sperm stimulation.

## 2. Materials and Methods

### 2.1. Gamete Collection, Oocyte Maturation, and Fertilization In Vitro

*Astropecten aranciacus* (starfish) were collected from the end of January to May in the Gulf of Naples and Gaeta and maintained at 16 °C in circulating seawater. Ovaries were isolated from the dorsal region of the arms of the animal, and the immature oocytes (GV-stage) were gently filtered with gauze and washed in natural seawater (NSW, pH 8.1), filtered with a Millipore membrane of 0.2 µm pore size (Nalgene vacuum filtration system, Rochester, NY, USA). The oocytes kept in NSW were used for maturation and fertilization experiments within the next 3 h. The dry sperm collected from the testis were diluted in NSW and used to inseminate oocytes and eggs at a final concentration of 1 × 10^6^ sperm/mL. The sperm density was calculated according to the following procedure: a piece of male gonad was surgically removed from the animal and kept in a 1.5 mL Eppendorf tube. A volume of 3.5 μL of the dry sperm was added to 1 mL of natural seawater (NSW) in an Eppendorf tube (Diluted Sperm Stock). To determine the sperm density in the Diluted Sperm Stock, 2 μL of aliquot was placed on the central groove of a hematocytometer, and then the number of sperm in a square millimeter was counted. The number obtained was 2602 (= N). This number was multiplied by 10 in order to obtaine the sperm count in 1 mm^3^, and then multiplied by 1000 to get the sperm count in 1 mL (Diluted Sperm Stock). Thus, the density of *A. aranciacus* dry sperm was around 0.7 × 10^10^ per mL. For fertilization experiments, 2 µL dry sperm were suspended in 1 mL NSW (500× dilution). From this solution, 10µL were added in 1 mL NSW containing the eggs (100× dilution). The final sperm concentration in the fertilization chamber would be 0.7 × 10^10^ per mL/50,000 (dilution ratio) = 1 × 10^6^ spermatozoa per mL. In vitro, maturation was performed by adding the hormone 1-methyladenine (1-MA) (Acros Organics, Fisher Scientific, Milan, Italy) to the GV-stage oocytes suspended in NSW at a final concentration of 10 μM. Isolated GV-stage oocytes placed in a small Petri dish were treated with DTT (at a concentration of 10 mM in NSW, pH 7.57) for 10 min, and the rate of sperm entry was judged by blebs formation under a light microscope. The extent of the maturation progress was monitored 70 min after incubating with 1-MA or DTT by examining the germinal vesicle breakdown (GVBD). To assess the effect of DTT after maturation, some oocytes were stimulated with 1-MA for 60 min and then exposed to DTT for 10 min. These eggs were inseminated with or without washing in NSW for comparisons. 

### 2.2. Light Microscopy and Transmission Electron Microscopy (TEM)

Light and electron microscopy observations were performed to monitor the surface and cortical changes in the oocytes before and after maturation. Whereas light microscopy was performed on live cells, transmission electron microscopy (TEM) samples were fixed in NSW containing 0.5% glutaraldehyde (pH 8.1) for 1 h at room temperature. After extensive washing in NSW, the specimens were post-fixed with 1% osmium tetroxide and 0.8% K_3_Fe(CN)_6_ for 1 h at 4 °C. The samples were washed in NSW, rinsed with distilled water (3 times, 10 min each), and finally treated with 0.15% tannic acid for 1 min at room temperature. The specimens were then dehydrated in ethanol with increasing concentrations. Residual ethanol was removed with propylene oxide before embedding in EPON 812. Ultrathin sections were made with the ultramicrotome Leica EM UC7 (Leica Microsystems, Wetzlar, Germany) and observed under a Zeiss LEO 912 AB (Carl Zeiss Microscopy Deutschland GmbH, Oberkochen, Germany) without staining.

### 2.3. Chemicals and Reagents

DL-Dithiothreitol (DTT) (Sigma-Aldrich, St. Louis, MO, USA) and Tris-(2-carboxyethyl)phosphine hydrochloride (TCEP) (ThermoFisher, Waltham, MA, USA) were dissolved in distilled water and used for bath incubation at the concentrations specified in the text. Hoechst-33342 and all other unspecified materials were purchased from Sigma Aldrich. 

### 2.4. Microinjection, Ca^2+^ Imaging, Fluorescent Labeling of F-Actin, and Jelly Coat

Intact immature and maturing oocytes were microinjected using an air pressure transjector (Eppendorf Femto-Jet, Hamburg, Germany). To monitor intracellular Ca^2+^ level changes following the insemination of GV-stage or maturing oocytes, immature oocytes were microinjected with 500 µM of Calcium Green 488 conjugated with 10 kDa dextran mixed with 35 µM Rhodamine Red (Molecular Probes, Eugene, OR, USA) in the injection buffer (10 mM Hepes, 0.1 M potassium aspartate, pH 7.0). The values of Ca^2+^ signals were obtained from three independent experiments (N) using three different females, and the number of eggs (n) being analyzed for each condition is specified in Section 3.

The fluorescence images of cytosolic Ca^2+^ changes following the insemination of immature and maturing starfish oocytes, whose number for each condition (n) is specified in the Results, collected from different females in three independent experiments were captured with a cooled CCD camera (Micro-Max, Princeton Instruments Inc., Trenton, NJ, USA) mounted on a Zeiss Axiovert 200 with a Plan-Neofluar 40/0.75 objective at about 2 s intervals. The data were analyzed with MetaMorph (Universal Imaging Corporation, Molecular Devices, LLC, San Jose, CA, USA). Following the formula, F_rel_ = [F − F_0_]/F_0_, where F represents the average fluorescence level of the entire egg and F_0_ the baseline fluorescence, the overall Ca^2+^ signals were quantified for each moment, and F_rel_ was expressed as RFU (relative fluorescence unit) for plotting the Ca^2+^ trajectories. Applying the formula F_inst_ = [F_t_ − F_(t−1)_]/F_(t−1)_, the instantaneous increment of the Ca^2+^ level was analyzed to locate the specific area of the transient Ca^2+^ increase. 

To visualize the actin cytoskeleton, 50 µM (pipette concentration in methanol) of the fluorescent F-actin probe AlexaFluor 568-phalloidin (Molecular Probes, Thermo Fisher Scientific, Hillsboro, OR, USA) was microinjected into the oocytes before and after maturation. The oocytes were then incubated with 1-MA or DTT to assess the structural changes in the actin cytoskeleton. Two independent experiments (N = 2) were performed utilizing oocytes from two different females. The fluorescence signals from the living cells in the imaging chambers were detected using a Leica TCS SP8X confocal laser scanning microscope equipped with a white light laser and hybrid detectors in the lightning deconvolution mode (Leica Microsystem, Wetzlar, Germany). The number of eggs examined for each condition (n) is specified in the Results.

Living immature oocytes induced to undergo maturation by 70 min of stimulation with 1-MA or DTT were incubated with 25 µM of the synthetic fluorescent polyamine BPA-C8-Cy5 [17] to label the jelly coat (Appendix A).

### 2.5. Visualization of Sperm inside Treated Oocytes before and after Maturation

Diluted sperm were stained with 5 µM of Hoechst-33342 (Sigma-Aldrich, Saint Louis, MO, USA) for 30 s before insemination. The labeled sperm nuclei incorporated into *A. aranciacus* immature oocytes and after treatment with 1-MA, DTT, or both eggs were counted in the cytoplasm 10–15 min after insemination using a cooled CCD (Charge-Coupled Device) camera (MicroMax, Princeton Instruments Inc., Trenton, NJ, USA) mounted on a Zeiss Axiovert 200 microscope with a Plan-Neofluar 40X/0.75 objective with a UV fluorescent lamp. The number of fertilized eggs examined (n) for each condition in three or four independent experiments (N) using three or four different females is shown in Section 3 and Tables.

### 2.6. Statistical Analysis

The numerical MetaMorph data were analyzed with Excel (Microsoft Office 2010) and reported as mean ± standard deviation (SD) in all cases in this manuscript. One-way ANOVA was performed with Prism Windows 5.04 (GraphPad Software), and *p* < 0.05 was considered statistically significant. For ANOVA results showing *p* < 0.05, the statistical significance of the difference between the two groups was assessed using Tukey’s post hoc tests. The two groups of data showing significant differences from each other were marked with symbols indicating the *p* values in the figure legends. The same experiment was repeated 2–4 times (N) utilizing the oocytes from as many animals. Thus, N refers to the number of replicates and n refers to the number of oocytes. On the other hand, the numerical data for the Ca^2+^ signal and different values were pooled together for statistical analysis by using all the oocytes treated similarly.

## 3. Results

### 3.1. The Effect of the Disulfide-Reducing Agent DTT on the Fertilization Response of A. aranciacus Immature Oocytes 

Immature oocytes isolated from the starfish ovary in normal seawater (NSW, pH 8.1) are polyspermic upon insemination due to the structural organization of their surface and cortex, allowing the fusion of multiple sperm with the oocyte plasma membrane [22,23]. The light microscopy image of Figure 1A shows that following the insemination of immature (GV-stage) oocytes in NSW, the fusion between the two gametes promotes the formation of several enlarged fertilization cones to incorporate many sperm into the oocyte cytoplasm (arrowheads). Occasionally, it is possible to observe the separation of VL from the oocyte plasma membrane induced by the exocytosis of cortical granules (CGs) only in regions of the oocyte cortex where CGs are positioned close enough to the plasma membrane and prompt fusion with it (arrows). The treatment of immature starfish oocytes with 10 mM DTT (pH 7.56 for 10 min) had several effects on the physiological response usually occurring at insemination. Specifically, a morphological sign of the reduction in the number of sperm entries is indicated by the formation of fewer fertilization cones on the surface of GV-stage oocytes following the insemination in the NSW containing DTT (Figure 1B) and after washing the disulfide-reducing agent. The light microscopy image in Figure 1A also shows a partial separation of the VL from the oocyte plasma membrane near the fertilization cone of the inseminated oocytes in NSW (arrows). When immature oocytes are incubated and inseminated in the presence of DTT, such separation does not take place (Figure 1B) even after washing off DTT. As shown in Figure 1F, the DTT treatment slightly alleviated the extent of polyspermy in the immature oocytes after insemination, as the number of sperm entries reduced to 6.32 ± 1.27 (N = 3, n = 80), which was significantly lower than the control oocytes inseminated in NSW (8.95 ± 198, N = 3, n = 80, *p* < 0.01). When the DTT was washed off prior to insemination, the number of sperm entries was reduced even higher (4.45 ± 1.37, n = 60, *p* < 0.01). 

The transmission electron microscopy (TEM) micrograph in Figure 2A shows the microvillar morphology of a GV-stage oocyte isolated from the ovary in NSW (control). The micrograph shows regions of the oocyte surface at lower and higher magnification (inset) where longer microvilli are visible among those embedded in the VL. The micrograph of the GV-stage oocyte surface following insemination (Figure 1C) depicts one of the many protrusions (the fertilization cone, arrow, and Figure 1A) that are induced by the fusion of multiple sperm with the oocyte plasma membrane beneath the partially separated vitelline layer becoming fertilization envelope (FE) that are easily detectable under the light microscope (Figure 1A). DTT treatment (10 min) induced striking changes in microvillar morphology, which is less evident in the region of the oocyte cortex containing longer microvilli that had lost their interdigitation profile (Figure 2B, inset). The electron micrographs also showed an alteration in the VL structure, such as ruptures of the modestly elevating FE (Figure 2D, arrows), which is appreciable only at the TEM level but not detectable under a light microscope (Figure 1B). DTT incubation also altered the organization of the content of the electron-dense material of the cortical granules (CGs), which lacked the usual round shape when it was released in the perivitelline space (PS), as seen in control experimental conditions (Figure 2D, arrowhead). 

### 3.2. DTT Treatment of Immature Starfish Oocytes Affects the Distribution of the Cortical F-Actin and Its Dynamics upon Insemination

Immature *A. aranciacus* oocytes were microinjected with Alexa 568-phalloidin to visualize in vivo the cortical and cytoplasmic F-actin distribution and its changes after sperm addition. The image of confocal laser scanning microscopy in Figure 3A shows the canonical network-like distribution of the actin filaments in the cytoplasm of immature oocytes (n = 8). Note the fluorescent actin indicator is excluded from the germinal vesicle (GV, nucleus). At the oocyte surface, two cortical F-actin layers (an external and internal one) are visible, which are rearranged in the regions of sperm entry (Figure 3A). About 5 min after insemination (Figure 3A′), two fertilization cones are shown to protrude externally to the oocyte surface (arrow), marking polyspermic fertilization, typical of the oocyte at this maturation stage. By 20 min after insemination (Figure 3A″), polymerized F-actin fibers are perpendicularly formed to the oocyte surface to incorporate the sperm (arrowhead). The effect of brief DTT treatment (10 mM, 10 min) on the structural organization of the cortical F-actin of immature oocytes before insemination and after sperm addition in the presence or absence of DTT is visible in the confocal images of Figure 3B. Before insemination, the incubation of the GV-stage oocytes with 10 mM DTT for 10 min induced further separation of the inner layer of F-actin from the outer one in the ectoplasmic region of the oocyte cortex (Figure 3B, n = 7). Moreover, the actin filaments surrounding the GV were partially detached from the GV nuclear envelope, evident in five out of seven oocytes (Figure 3B, arrow). After sperm addition, activated oocytes can still experience the polymerization of F-actin needed to incorporate the multiple sperm (arrowhead). Still, the cytoskeletal organization was slightly altered 5 and 20 min later compared to the control oocytes (Figure 3B′,B″). Even if the DTT washing restored the distribution of the cortical F-actin and slightly eased the structural reorganization of the F-actin in the oocyte cortex to incorporate the sperm 20 min after insemination (Figure 3C″, arrowhead, n = 7), the fertilization response of DTT-treated and washed oocytes remained altered, as indicated by the reduction in sperm entry shown in the fluorescent images and histograms of Figure 1 and the single fertilization cone depicted in Figure 3C′ (arrow). 

### 3.3. DTT Treatment Does Not Induce Germinal Vesicle Breakdown (GVBD) and Maturation in A. aranciacus Starfish Oocytes

Since it has been known that the reducing agent DTT induces germinal vesicle breakdown (GVBD) and maturation in the oocytes of *Asterina pectinifera* [29,30,32], it was tested if the same was true in another species of starfish. To this end, the GV-stage oocytes of *A. aranciacus* were exposed to 10 mM DTT for 70 min. The treatment of the oocytes of this Mediterranean starfish species with the maturing hormone 1-MA represents the length of time during which optimal fertilizability conditions are acquired. After this precise maturation time, the eggs experience a standard fertilization response comprising the F-actin-dependent events, i.e., the sperm-induced Ca^2+^ signals, exocytosis of the cortical granules, cortical cytoplasmic reorganization, and entry of only one sperm [11]. Whether incubation with DTT promotes meiotic maturation in the GV-stage oocytes of *A. aranciacus* was examined by looking at the occurrence of GVBD, which is the morphological sign that the oocytes have resumed the meiotic cycle. Our results from three different animals (N = 3) showed that 70 min of DTT treatment did not induce GVBD, as some nuclear components are still visible in the cytoplasm. This observation is at variance with the previous reports that > 5 mM DTT induces GVBD in starfish oocytes [29,30,31,32,33]. Possible explanations for this discrepancy will be addressed in the Discussion (see below). The light microscopy images in Figure 4 represent the cortical reaction (5 min after fertilization) displayed by *A. aranciacus* oocytes (GV-stage) that were inseminated after various pretreatments: (A) matured with 1-MA for 70 min, (B) incubation with DTT for 70 min, (C) matured with 1-MA for 60 min and then exposed to DTT for 10 min. In Figure 4A, the oocyte matured by 1-MA stimulation exhibited a clear sign of GVBD, as indicated by the complete intermixing of the nucleoplasm with the cytoplasm. At fertilization, these eggs underwent the total and equidistant elevation of the fertilization envelope (FE) due to the exocytosis of CGs and cortical F-actin polymerization in the perivitelline space. By contrast, the DTT treatment did not result in GVBD, as is evidenced by the presence of well-delineated GV, which is still visible in the cytoplasm of the DTT-treated and fertilized *A. aranciacus* oocytes after washing (Figure 4B). Another morphological effect of DTT treatment is the impairment of the elevation of the FE judged by light microscopy observations compared to the control. A shorter treatment of DTT (10 min) following 1-MA-induced GVBD following 60 min of stimulation was sufficient to affect the structural integrity of the VL of the unfertilized DTT-treated oocytes, leading to an inhomogeneous and not distinct separation of the FE from the cell plasma membrane (Figure 4C). Upon light microscopy, washing off the DTT before insemination restored the structural integrity of the VL of the unfertilized oocytes, which underwent CG exocytosis and an elevation of the FE upon insemination, similar to the control. 

The analysis of the sperm entry in the eggs with or without subsequent treatment with DTT showed that they were predominantly penetrated by only one sperm (Figure 4D–F and histograms in G). Specifically, the 1-MA-stimulated oocytes (control) were penetrated by only one sperm (N = 2 and n = 40 eggs, 1 ± 0). The treatment of immature oocytes with DTT for 70 min produced only three polyspermic zygotes out of 35 fertilized eggs, and the rest were all monospermic (n = 35 eggs, 1.11 ± 0.16). The mature eggs exposed to DTT (60 min of 1-MA and DTT for 10 min) were also primarily monospermic, except for one polyspermic zygote (n = 40 eggs, 1.1 ± 0.19). Washing off DTT before insemination did not make much difference because most of the zygotes were monospermic except for three (n = 40, 1.02 ± 0.08). The examination of the early-stage embryonic development of the immature oocytes treated as described above (N = 3 females, n = 100 oocytes for each experimental condition) has shown that, at variance with the 100% normal cleavage in control, the percentage of cleavage inhibition following the fertilization of immature oocytes treated with DTT for 70 min was 95%. The monospermic fertilization following the external application of 1-MA (60 min) and DTT (10 min) to the immature oocytes produced 31% and 23% of cleavage inhibition when insemination was performed after washing DTT from seawater, respectively (Appendix A).

### 3.4. DTT Treatment Affects the F-Actin Rearrangement during Oocyte Maturation and Fertilization

The stimulation of *A. aranciacus* immature oocytes with 1-MA promotes a dramatic structural reorganization of the F-actin in the cortex accompanied by GVBD, which is essential for a successful fertilization response [16,28]. Since the “maturation” of oocytes with DTT compromised the cortical reaction (Figure 4B), we tested the effect of long and short DTT treatment on the dynamic changes in the cortical F-actin at fertilization when immature oocytes were stimulated by 1-MA for 70 min (control), by DTT treatment (for the same time), or by a combined treatment: with 1-MA for 60 min to ensure GVBD and then with DTT for 10 min with and without DTT in NSW. The laser scanning microscopy images in Figure 5 show the F-actin staining in living oocytes matured with 1-MA, DTT, or both (A–D). The microinjected AlexaFluor 568-phalloidin (50 µM in the pipette) visualized actin fibers oriented perpendicularly to the surface in the cortical region of an egg matured with 1-MA for 70 min (Figure 5A, n = 13) [12]. By contrast, the same amount of the F-actin probe in the oocytes treated with DTT for 70 min evidences a remarkable signal reduction in the cortex (Figure 5B, n = 13). The GV is still visible in the cytoplasm, implying that GVBD did not occur in the oocytes treated with DTT. On the other hand, brief (10 min) DTT exposure of the oocytes that were already matured by 1-MA (60 min) compromised the reorganization of the F-cortical actin. The image of Figure 5C shows the discontinuity of the cortical layer of F-actin induced by the short DTT treatment (n = 10), which did not recover its structure after removing DTT from seawater (Figure 5D, n = 11). The effect of DTT on the cortical F-actin remodeling during oocyte maturation was more evident after insemination. The overlay image of Figure 5A′ shows the formation of the fertilization cone on the surface of an oocyte matured by 1-MA (control) comprising the structures of cytoplasmic actin filaments that are polymerized to incorporate the DNA-stained sperm (blue spot and arrow) by 5 min after insemination. About 20 min later, the sperm (blue spot and arrow) was guided more profoundly into the cytoplasm by the actin filaments (Figure 5A″). In the oocytes treated for 70 min with DTT, by 5 min after insemination, the reorganization of the F-cortical structure (Figure 5B) was heavily compromised with respect to the cortical actin polymerization that incorporated the sperm (Figure 5B′). Only 20 min after insemination, a sort of polymerization of F-actin structures was visible (albeit with a disarranged morphology) in the cortex of the zygotes (Figure 5B″). The incubation of the oocytes with DTT (10 min) after their stimulation with 1-MA for 60 min was sufficient to alter the formation of the fertilization cone upon insemination (arrow), the actin filament translocation towards the center of the zygote, and the polymerization of the cytoplasmic F-actin structures to guide the sperm inside the egg, which is usually visible 5 and 20 min after insemination (Figure 5C′,C″). This DTT-induced F-actin deregulation was not much reversed if insemination was performed after washing off the DTT, as shown in the images taken 5 and 20 min after insemination (Figure 5D′,D″).

### 3.5. Ultrastructural Changes in the Vitelline Layer and Microvillar Morphology Following DTT Treatment

Observations from the transmission electron microscope (TEM) showed that the ultrastructure of the oocytes stimulated with DTT for 70 min did not mimic, but was strikingly different from, that of the *A. aranciacus* eggs that were normally matured by 1-MA. The micrograph in Figure 6A shows the cortical granules (CGs), after 70 min of maturation with 1-MA, oriented perpendicularly to the surface beneath the microvilli that are embedded in the thick vitelline layer (VL) (Figure 6A). The immature oocytes stimulated with DTT for the same length of time promoted a striking alteration in the vitelline layer’s structure, whose increased electron density made the microvilli morphology difficult to visualize (Figure 6B, arrow). Furthermore, the CGs in the DTT-treated oocyte were more often filled with electron-dense materials than the control oocytes matured by 1-MA. Remarkably, filaments are visible in the remaining part of the CGs (arrow) due to the action of DTT (Figure 6B). In this regard, it is worth noting that immature oocytes were subjected simultaneously to the exact fixation protocol, and the ultrathin sections were not stained as described in Materials and Methods. Washing off DTT before fixation of the samples reduces its effect on the structural organization of the VL, as evidenced by the less electron-dense staining of the envelope, allowing the visualization of the embedded microvilli (Figure 6C). The TEM analysis on the cortical reaction of *A. aranciacus* oocytes fertilized after stimulation with 1-MA (control) is shown in the micrograph of Figure 6A′. Five minutes after insemination, normal CG exocytosis was highlighted by the absence of CGs on the subplasmalemmal region of the fertilized egg. The image shows the electron-dense content of the CGs released in the perivitelline space (PS), with the standard round shape (arrowhead), which will then fuse with the inner face of the newly formed fertilization envelope from the VL (arrowhead). Microvilli elongation is also easily visible. The effect of DTT treatment on the structure of the VL of oocytes is manifested by the altered integrity of the FE 5 min after insemination (Figure 6B′), as judged by the protrusion of the elongated microvilli from the FE due to its partial destruction (arrow). The effect of DTT on the structure of the CGs in the oocytes exposed to the reducing agent for 70 min and fertilized is also illustrated. At variance with the control, in the DTT-treated oocytes, the electron-dense content of the cortical granules secreted into the PS (Figure 6B′, arrowheads) shows a CG morphology different from that produced in the control samples in Figure 6A′. The partial destruction of the FE was not reversed by washing off the DTT before insemination. The altered morphology of the secreted content of the CGs in the PS is similar to that released following the insemination of immature GV-stage oocytes shown in Figure 2D (arrowheads). 

### 3.6. Altered Sperm-Induced Ca^2+^ Response in the DTT-Treated Oocytes of Starfish

Previous studies on maturing oocytes of *A. aranciacus* have shown that the F-actin-based structural remodeling of microvilli and the cortex plays a crucial role in regulating the pattern of the Ca^2+^ response at fertilization [13,16,28]. The finding that a brief treatment (10 min) of immature oocytes with DTT (10 mM) affects the structural dynamic of the F-actin and sperm entries prompted us to investigate the effect of the cytoskeletal change on Ca^2+^ responses at fertilization. The pseudocolor images of the Ca^2+^ changes following the insemination of a control immature oocyte after 10 min of incubation in NSW (Figure 7A) show that out of 16 immature oocytes (N = 3, n = 16), 12 responded to insemination with fewer Ca^2+^ spots (4.56 ± 0.87, Table 1), as a result of the interaction and fusion of multiple sperm (arrowheads). The subsequent cortical Ca^2+^ flash (CF) reaching an average peak amplitude of 0.166 ± 0.010 RFU was followed by numerous Ca^2+^ waves (CWs) (0.59 ± 0.011 RFU) originating from multiple cortical Ca^2+^ spots (Figure 7A and the green lines in the graphs in D). In the immature oocytes treated with DTT for 10 min (n = 16), the reduction in the Ca^2+^ spots (3.07 ± 0.6, *p* < 0.01) with respect to the control (Figure 7B) is in line with the reduced rate of polyspermy shown in Figure 1F. The CF recorded in 12 out of 16 oocytes reached an average Ca^2+^ peak amplitude (0.047 ± 0.07 RFU), and the CW (0.31 ± 0.09 RFU) was significantly lower than those in the control oocytes (*p* < 0.01) (Figure 7B, see the purple lines in the graphs in D). Insemination after washing off the DTT following 10 min of 1-MA treatment reduced the number of cortical Ca^2+^ spots (2.63 ± 0.75, *p* < 0.01) (Table 1); i.e., fewer sperm could fuse with the oocyte plasma membrane. This is also indicated by the single CW running in the oocyte, originating from the Ca^2+^ spot induced by the fusion between the two gametes (Figure 7C, arrowhead and blue lines in the graphs in D). The CW reached an average Ca^2+^ peak of 0.26 ± 0.04 RFU, *p* < 0.01, significantly lower than that originating from the confluence of multiple waves generated in the experimental conditions shown in A and B. No differences were found in the time required to record the first Ca^2+^ signal in the control (time of activation) and DTT-treated oocytes after insemination (Table 1).

The effect of DTT on F-actin structural dynamics during oocyte maturation is reflected in the pattern of the Ca^2+^ response upon insemination. The sperm-induced Ca^2+^ response of the physiologically stimulated oocytes with 10 μM 1-MA for 70 min is shown in Figure 8A. The first Ca^2+^ signal in these eggs was detected 34.15 ± 8.92 sec after insemination (time of activation, Table 2). The cortical flash (CF) was followed by a Ca^2+^ wave (CW) with a lag of a few seconds and propagated from the sperm–egg fusion site to the opposite pole (traverse time, 129.9 ± 12.05 s, Table 2). The graphs (green lines in Figure 8E), representative of 38 control mature eggs (n = 38), show the average Ca^2+^ peak amplitudes of the CF (0.095 ± 0.03 RFU) and the CW (1.19 ± 0.13 RFU). The effect mentioned above, of DTT on F-actin dynamics and the Ca^2+^ response at fertilization, is evident in the pseudocolor image and graphs (purple lines) of Figure 8B. Upon insemination, only 1 out of 18 eggs (n = 18) responded to the sperm with a small CF (0.036 RFU) as compared to that in the control (Table 2). The CW measured in 9 out of 18 DTT-treated oocytes reached a lower Ca^2+^ peak (0.89 ± 0.25 RFU) in comparison with the normally matured oocytes (1.19 ± 0.13 RFU, *p* < 0.01). Notably, the time required to initiate the CF was much longer in the oocytes treated with DTT for 70 min (203.45 ± 97.6 s, *p* < 0.01) than in the control (34.15 ± 8.92 s), implying that DTT modified the nature of the jelly coat, which was much reduced on the egg surface as shown in Appendix A; it may thereby have lost its full functionality in inducing the F-actin-based long acrosomal process on the sperm head [11,16]. Ten minutes of incubation of the oocytes with DTT after treatment with 1-MA for 60 min was again sufficient to alter the Ca^2+^ response at fertilization without affecting sperm mobility (see the pseudocolor images and the graphs, brown lines in E, in Figure 8C) due to the depolymerizing effect of DTT on the cortical F-actin of the unfertilized treated oocytes (Figure 5B). The addition of sperm to the seawater containing DTT elicited a Ca^2+^ response in 24 out of 25 cells (n = 25). The Ca^2+^ peak of the CF (0.031 ± 0.01 RFU, *p* < 0.01) and CW (0.55 ± 0.19 RFU, *p* < 0.01) detected in 18 out of 25 oocytes matured with 1-MA and DTT showed amplitudes that were significantly lower than those experienced in the control (Table 2). No difference was observed in the time required to start the Ca^2+^ response (CF) after insemination (Table 2). Washing off the DTT before the insemination of oocytes stimulated with 1-MA for 60 min and an additional 10 min with DTT produced the occurrence of CF in 8 out of 18 cells (n = 18) (Figure 8D and graphs, orange lines in E). The absence of DTT in seawater during fertilization also slightly increased the Ca^2+^ peak amplitude of the CF (0.063 ± 0.02 RFU, *p* < 0.05) and CW (0.70 ± 0.09 RFU, *p* < 0.01).

## 4. Discussion

Our previous studies have shown that the normal starfish oocyte maturation process leading to GVBD and a normal fertilization response strictly depends on the reorganization of the surface and cortical F-actin of the immature oocytes. The maturation hormone 1-MA acting on the oocyte plasma membrane triggers actin remodeling and Ca^2+^ changes in the oocytes that can be easily monitored under experimental conditions that resemble those occurring in nature, i.e., using intact and not manipulated cells [11,12,51,52,53,54]. Such studies have clarified that in immature (GV-stage) starfish oocytes, the fusion of multiple sperm with the oocyte plasma membrane [20,22,23] is firmly dependent on the structural organization of the extracellular vitelline layer covering the oocyte plasma membrane [11,13,14,16], while the Ca^2+^ and electrophysiological changes after the insemination of immature or maturing oocytes are linked to the length of microvilli (containing actin filaments) and to the structure of the CGs positioned beneath the oocyte plasma membrane [8,10,55,56]. During the maturation of starfish oocytes, the F-actin-based reconstitution of the oocyte cortex (microvilli shortening and CG orientation) with the contribution of the nuclear components following GVBD establish the morpho-functional changes necessary to elicit appropriate electrical and Ca^2+^ responses upon sperm stimulation [5,11,12,13,16]. In line with this, the Ca^2+^ signals during the 1-MA-triggered cortical actin restructuration of starfish oocytes [9,13,53] are associated with cycles of decreasing and increasing in the rigidity (stiffness) of the maturing oocytes [31,57,58,59,60]. The rearrangement of the microvillar actin filaments that parallel the changes in potassium conductance in the regulation of the fertilization potential after the maturation process of starfish oocytes [55] could provide a means for the changes in the electrical properties of the plasma membrane of maturing oocytes and for controlling the opening of channels that are located on the tip of microvilli and the influx pathway for external Ca^2+^ into the cell [61,62,63]. The different sperm-elicited Ca^2+^ peak amplitude of the CF in immature and maturing oocytes (Figure 7 and Figure 8), endowed with longer and shorter microvilli, respectively (Figure 2 and Figure 6) [10,13,14], could be due to the Ca^2+^ release following the disassembly of the microvillar diffusion barrier system and Ca^2+^ influx, in line with the above microvillar Ca^2+^ signaling concept. 

Similarly, a close relationship between the microvilli morphology of sea urchin eggs and the pattern of the CF at fertilization has also been reported [11,64]. The results described in this work confirmed the crucial role played by the structural and dynamic changes in the cortical actin cytoskeleton during oocyte maturation for a normal fertilization response. They also highlighted the striking difference between the F-actin structural changes induced by the natural hormonal and artificial stimulation by DTT, since previous studies showed that >5 mM DTT caused oocyte maturation and GVBD in starfish oocytes including *Asterina pectinifera*. 

To elucidate the molecular mechanism by which 1-MA resumes the meiotic cycle in starfish oocytes, previous studies compared the increase in the sulfhydryl (SH) contents in the cortical proteins of the oocytes after the incubation with 1-MA or DTT. The results showed that both 1-MA and DTT treatment caused an increase in the amount of protein-SH in the oocytes, suggesting that a disulfide-reducing action of 1-MA is essential for resuming the maturation of *A. pectinifera* oocytes [29,30]. In this work, we studied the in vivo effect of DTT on immature and maturing *A. aranciacus* oocytes by looking at their fertilization response, since it is well known that the actin cytoskeletal changes in vitro, i.e., the depolymerization and polymerization states (whether G or F-actin) are influenced by the SH groups on cysteine, whose oxidation helps to form an intermolecular disulfide (S-S) bond to yield an actin dimer [65,66].

The results in Figure 1 and Figure 3 show that this was indeed the case. When immature *A. aranciacus* oocytes known to be polyspermic were treated for 10 min with DTT, the rate of sperm entry was reduced as a result of the alteration in the cortical actin of the oocyte induced by the disulfide-reducing agent. DTT treatment also affected the actin polymerization to form structures incorporating sperm in the cytoplasm (Figure 3). 

As for the “maturation process” induced by DTT, the results of the present study show that DTT treatment for 70 min did not induce GVBD in *A. aranciacus* oocytes. This result is at variance with the previous findings in other species of starfish. One explanation for this discrepancy may be that the induction of DTT in the larger oocytes (>250 μm in diameter) of *A. aranciacus* might require a much longer incubation time. Alternatively, the given experimental condition with 10 mM DTT, which was automatically buffered at pH 7.56, might be acidic enough (deviating from 8.1 of NSW) to inhibit the activity of some components comprising the pathway of the maturation-promoting factor (MPF) or the cyclin B-Cdk1 (Cdk1). For example, it has been shown in *A. pectinifera* that a reduced intracellular pH delays GVBD, even in the presence of high Cdk1 activity [67]. Thus, the negative data regarding the effect of DTT mimicking 1-MA to induce GVBD in *A. aranciacus* require further investigation. Nonetheless, a drastic DTT effect was evident at the level of the cortical region as the DTT-treated oocytes failed to undergo a full elevation of the FE after insemination when observed under a light microscope (Figure 4B). 

The difference from the results with *A. pectinifera* oocytes could be ascribed to the distinct morpho-functional features of the oocytes/eggs of the two starfish species. Indeed, it is worth mentioning that *A. pectinifera* oocytes respond to the same amount of the Ca^2+^-linked second messengers, InsP_3_ and NAADP, with Ca^2+^ responses that are higher than those elicited in *A. aranciacus.* By contrast, *A. pectinifera* oocytes are irresponsive to the action of cADPr [51,53,68]. 

As a membrane-permeant reducing agent, DTT is expected to disrupt disulfide bonds in the proteins and other macromolecules on the surface and in the cortex of the oocytes. The ultrastructural TEM analyses confirmed the action of DTT leading to a partial destruction of the vitelline layer (VL) and microvilli morphology changes in immature and maturing *A. aranciacus* oocytes (Figure 2B and Figure 6B,C), as initially shown in *A. pectinifera* oocytes [30]. The striking alteration in microvillar morphology (Figure 2B, inset) may explain the inability of the F-actin-based long acrosomal process of the sperm to fuse with the oocyte plasma membrane [13,14] after reaching the preferential sites, where microvilli are much longer due to their changed morphology (Figure 2B, inset). The structural alteration in the *A. aranciacus* oocyte surface by DTT also includes changes in the internal structure of the cortical granules (CGs), as evidenced by the enhancement in the electron density of the material composing the core of the CGs as well as the visualization of filament-like structures in the remaining part of the CGs after the treatment with DTT (Figure 6B,B′). The effect of DTT on the morpho-functionality of CGs is also expressed in the rosette-like and less compact electron-dense materials being extruded into the perivitelline space (PS), as opposed to the round form that is routinely found in control (arrowheads in Figure 2C,D and Figure 6A′–C′). 

The changes in microvilli morphology and the structure of the CGs are linked to the alteration in sperm-induced Ca^2+^ response in immature oocytes and oocytes stimulated with DTT treatments (Figure 7 and Figure 8). These findings add weight to our previous results in fertilized sea urchin eggs, demonstrating that microvillar actin filaments and CGs contribute to shaping the Ca^2+^ response following sperm stimulation [64,69,70,71,72]. 

A comparison between the fertilization responses in immature oocytes induced to undergo maturation via the external application of 1-MA or DTT confirmed the disulfide-reducing agent’s effect on the actin’s conformational changes in vivo. It also highlighted striking differences in the mode of action of natural and artificial maturing agents, as previously reported [30,31]. That is, while 1-MA exerts its action on the oocyte plasma membrane, DTT profoundly alters VL and microvilli morphology and, by diffusing through the oocyte plasma membrane, also affects the structure of the CGs localized beneath the plasma membrane. The effect of DTT requiring its action inside the oocyte was indicated by the results that another reducing reagent but membrane impermeant, namely TCEP [73], did not resume the maturation process, leaving GV intact into the cytoplasm at a concentration of 0.5, 2, and 10 mM when applied to immature oocytes. 

At variance with 1-MA stimulation, the cortical actin depolymerization induced by treating immature oocytes for 70 min with DTT without promoting GVBD (Figure 5B) compromised the sperm-induced Ca^2+^ response. It also inhibited its subsequent reorganization (polymerization) necessary to assemble the F-actin structures 5 and 20 min after insemination in the cortical region of the zygote to incorporate the sperm (Figure 5A′,A″,B′,B″ for a comparison with the control). 

As for the crucial role of the F-actin in controlling Ca^2+^ signaling in starfish oocytes, it may be worth mentioning the dramatic changes leading to a disintegration of the cortical actin, which is not followed by a rapid surface reconstitution when ionomycin and latrunculin-A (LAT-A) are used [54]. Previous results have shown that the intracellular Ca^2+^ increase after a brief exposure of *A. aranciacus* immature oocytes to the Ca^2+^ ionophore ionomycin is accompanied by rapid and irreversible ultrastructural changes in the cortex of the oocyte, which include microvilli retraction and the fusion of the cortical granules with other vesicles. The fact that ionomycin pretreatment of the immature oocytes disrupted the functionality of the cortical actin cytoskeleton was evidenced by the compromised Ca^2+^ release, CG exocytosis, and development after their physiological maturation with 1-MA [74]. Similarly, the artificial activation of mature *A. aranciacus* eggs with LAT-A, in addition to promoting an altered Ca^2+^ response (oscillations and not a single Ca^2+^ transient as experienced by the fertilized eggs), inhibited the subsequent microvilli elongation following the cortical actin polymerization [75].

The abnormal fertilization response elicited by the artificial maturation induced by DTT highlights the independent irritability of the cortical F-actin and the importance of stimulating it physiologically [21]. Indeed, the drastic depolymerization (disintegration) of the ectoplasmic actin induced by DTT treatment by destroying its vital activity prevents reconstitution (polymerization), which would have followed biological stimulation [21,76].

## 5. Conclusions

The exposure of *A. aranciacus* oocytes to 1-MA for 70 min is required for GVBD, the intermixing of the nucleoplasm with the cytoplasm, and the restructuration of the cortical actin cytoskeleton with the contribution of nuclear components. Following oocyte maturation, the structural changes in F-actin allow the oocyte to mature into eggs to respond to sperm with a normal fertilization reaction, including a proper Ca^2+^ response, cortical granule exocytosis, and cleavage. Since disulfide-reducing agents such as dithiothreitol (DTT) are known to induce the maturation and GVBD of the oocytes of other starfish species, we analyzed the pattern of the fertilization response of *A. aranciacus* oocytes matured with 1-MA or after “maturation” induced by DTT. Even if actin is one of the proteins that DTT acts on, its drastic depolymerizing effect on the actin cytoskeleton of the oocytes deregulated the formation of the F-actin structures properly incorporating the sperm. The results of the morpho-functionality analysis of the cortical F-actin in living oocytes matured with 1-MA or DTT and inseminated revealed striking differences between the mode of action of the physiological stimulation by the natural hormone 1-MA and that induced by the artificial agent DTT. Our results show the crucial role of the changes in the state of cortical actin in living cells induced by physiological stimulation (hormone and sperm) to regulate these two biological functions. Further studies can shed light on the differences between actin dynamics, mainly studied under in vitro conditions, with conformational state changes exerted by actin in physiological conditions.

## Figures and Tables

**Figure 1 biomolecules-13-01659-f001:**
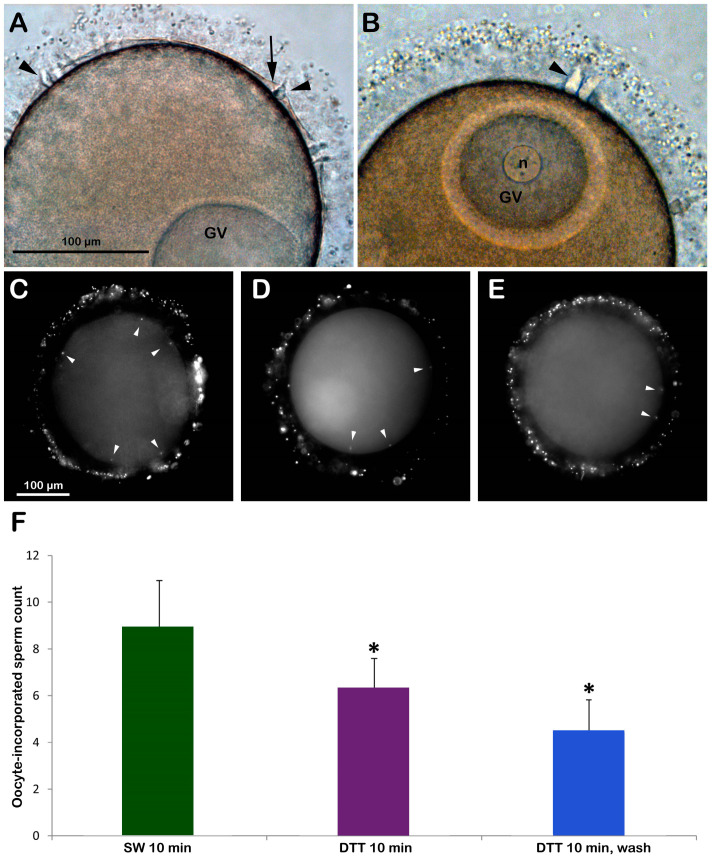
Brief exposure to DTT before insemination reduces the extent of polyspermy in immature (GV-stage) oocytes of *A. aranciacus.* (**A**) Insemination of GV-stage oocytes suspended in NSW induces the formation of multiple fertilization cones (arrowheads) due to polyspermic fertilization. Note the localized separation of the fertilization envelope (arrow) in some restricted surface areas. (**B**) DTT treatment (10 min) reduces the number of fertilization cones (arrowhead) as fewer sperm activate and penetrate the oocytes. Oocyte-incorporated sperm (arrowheads) were visualized via the fluorescent dye binding to DNA (**C**–**E**) and were counted to be presented in the histograms (**F**). Epifluorescence photomicrographs of representative oocytes inseminated in NSW with (**D**) or without DTT exposure (**C**) were compared with the ones inseminated after DTT exposure (10 min) and washed with NSW (**E**). * Tukey’s post hoc test, *p* < 0.01.

**Figure 2 biomolecules-13-01659-f002:**
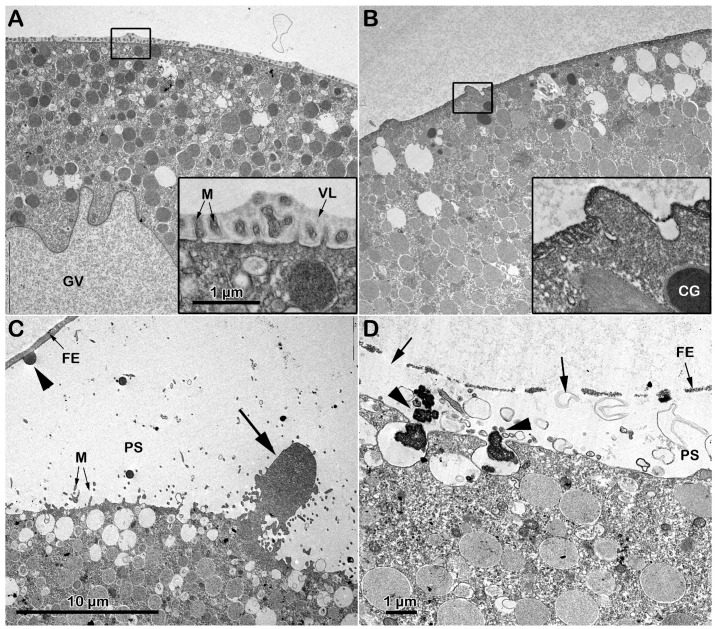
Effect of DTT on the surface of *A. aranciacus* immature oocytes before and after insemination. (**A**) Ultrastructure of a control GV-stage oocyte in NSW visualized with transmission electron microscopy (TEM) at lower and higher magnification (inset). The vitelline layer (VL) appears as a homogenous layer in which microvilli (M) are orderly embedded. (**B**) A brief DTT treatment (10 min) of the oocytes induces changes in the VL structure and microvillar morphology, which is more evident in the regions where cytoplasm bulges next to the microvilli exhibiting interdigitation profile (inset). (**C**) The TEM micrograph of the immature oocytes inseminated in NSW depicts a fertilization cone (arrow) formed beneath partially elevating FE due to CG exocytosis (arrowhead) and microvilli (M) elongation. (**D**) Insemination of DTT-treated oocytes evidences partial destruction of the VL (arrows), which now forms the fertilization envelope (FE), and more frequent occurrence of the electron-dense content within the CGs being released in the perivitelline space (PS) (arrowheads).

**Figure 3 biomolecules-13-01659-f003:**
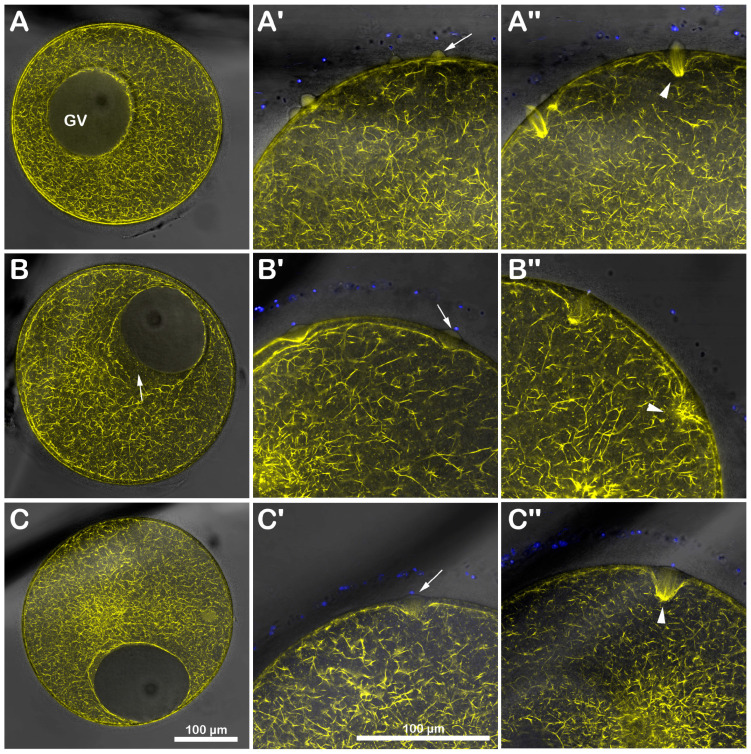
DTT-induced changes in the cortical F-actin distribution and dynamic before and after insemination of starfish immature oocytes. Live *A. aranciacus* GV-stage oocytes injected with Alexa 568-phalloidin were treated with DTT and inseminated to monitor the real-time changes in the F-actin with confocal laser scanning microscopy. Network of F-actin distribution in control oocytes in NSW before (**A**) and after insemination: 5 min (**A′**) and 20 min (**A″**). The overlay in (**A′**) with images of fluorescently labeled sperm head (blue) shows the formation of multiple fertilization cones (arrow) with polymerization of actin fibers to incorporate the sperm (**A″**, arrowhead). (**B**) DTT treatment (10 mM, 10 min) induces alteration in the outermost cortical F-actin structure (note the partially detaching of actin filaments from the GV nuclear envelope indicated by an arrow), which compromises the subsequent formation of the actin fibers (arrowhead) incorporating sperm 5 min (**B′**) and sometimes even 20 min after insemination (**B″**). This tendency is evident in DTT-treated oocytes inseminated after washing in NSW (**C**). The less defined cortical polymerization of actin filaments (arrowhead) at 5 min (**C′**) and 20 min (**C″**) may explain why the oocytes are penetrated by fewer sperm, as shown in Figure 1.

**Figure 4 biomolecules-13-01659-f004:**
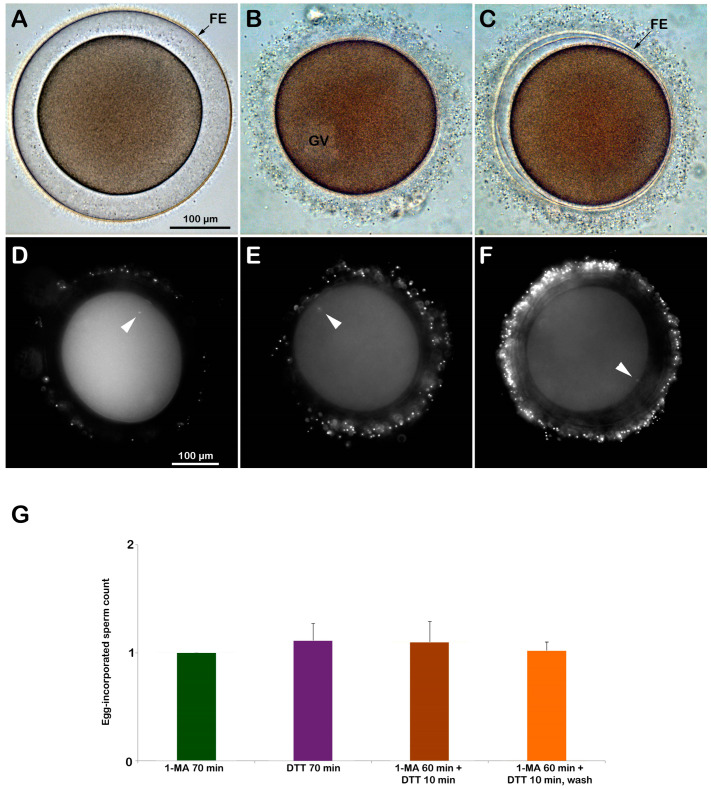
DTT exposure does not induce GVBD in immature *A. aranciacus* oocytes and alters their cortical reaction upon insemination. (**A**) A GV-stage oocyte stimulated with the natural maturing hormone 1-MA underwent GVBD by 70 min incubation, displaying the total and equidistant elevation of the fertilization envelope (FE) 5 min upon insemination. (**B**) A GV-stage oocyte treated for 70 min with DTT failed to undergo GVBD and intermixing of the nucleoplasm with the cytoplasm. Five minutes after insemination, these treated oocytes did not display a clear sign of FE elevation under the light microscope. (**C**) DTT (10 min) added after the incubation of immature oocytes with 1-MA for 60 min was sufficient to compromise the exocytosis of the CGs and the subsequent elevation of the fertilization envelope (FE), despite the apparently normal proceedings of GVBD. The number of DNA-stained sperm in the zygotes was assessed 10 min after insemination by counting the fluorescent signals corresponding to the sperm head (arrowheads) in oocytes stimulated for 70 min with 1-MA (**D**), oocytes treated for 70 min with DTT (**E**), oocytes stimulated for 60 min with 1-MA and then treated with DTT for 10 min (**F**). The results are presented in histograms (**G**). Predominantly, only one sperm entered (monospermic) in all the described experimental conditions.

**Figure 5 biomolecules-13-01659-f005:**
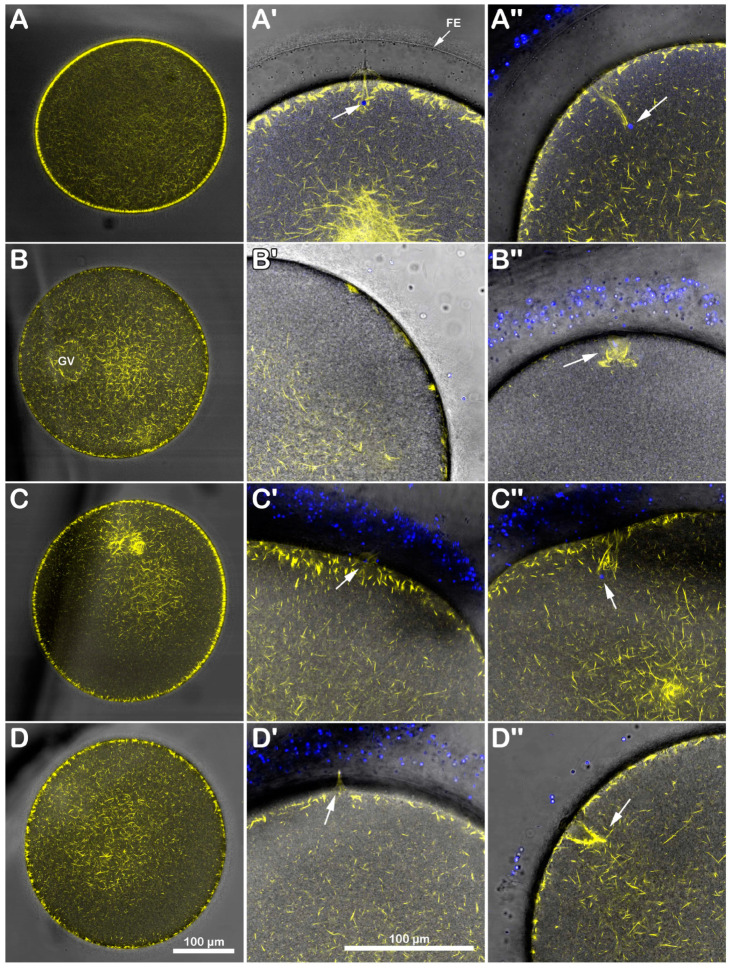
DTT treatment interferes with F-actin dynamics in maturing oocytes. *A. aranciacus* immature oocytes pre-injected with Alexa 568-phalloidin were stimulated for 70 min with 1-MA, and the changes in F-actin distribution were monitored before (**A**) and after insemination with confocal laser scanning microscopy: 5 min (**A′**); 20 min (**A″**). Note that actin filaments oriented perpendicularly to the plasma membrane of the unfertilized mature egg. The overlay image in (**A′**) indicates that sperm addition promotes the elevation of the fertilization envelope (FE) and cortical F-actin remodeling along the fertilization cone to capture the sperm about 5 min after insemination (blue spot and arrow) and more profoundly in the cytoplasm 20 min later (**A″**, blue spot and arrow). (**B**) Treatment of an immature oocyte for 70 min with DTT does not induce GVBD but reorganizes the cortical actin cytoskeleton in a way that is distinct from what 1-MA does. As the cortical actin cytoskeleton is more depolymerized in these eggs, the subsequent insemination does not produce the fertilization cone formation by 5 min (**B′**). Only 20 min after sperm addition is it possible to detect actin fibers in the cytoplasm with an altered structural organization (**B″**, arrow). (**C**) The cortical F-actin remodeling in *A. aranciacus* oocytes stimulated for 60 min with 1-MA to induce GVBD and then exposed to DTT for 10 min. The effect of the short external application of DTT is manifested in the impairment of the cortical actin dynamics following fertilization in the presence of DTT, leading to the formation of a morphologically altered fertilization cone and F-actin structure to incorporate the sperm 5 min and 20 min after insemination (blue spots and arrows in (**C′**) and (**C″**), respectively). (**D**) The cortical actin distribution in an oocyte washed with NSW after being incubated for 60 min with 1-MA and then with DTT for 10 min. Insemination in NSW still affects the cortical actin dynamics to form the fertilization cone and to bring in the sperm (arrows) by 5 min (**D′**) and 20 min (**D″**) after insemination.

**Figure 6 biomolecules-13-01659-f006:**
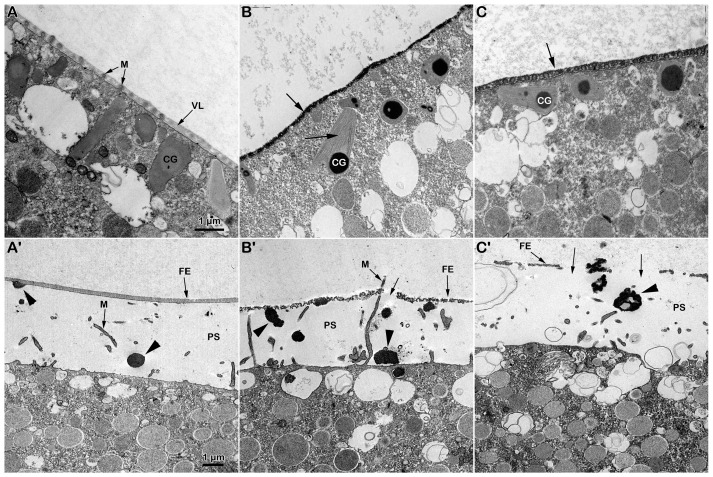
Ultrastructural analyses of DTT-treated immature oocytes before and after insemination. (**A**) TEM micrograph of an *A. aranciacus* GV-stage oocyte stimulated with 1-MA for 70 min. The microvilli (M) are ensheathed by the vitelline layer (VL). Beneath that, cortical granules (CGs) are positioned perpendicularly to the mature egg surface. (**A′**) Five minutes after insemination, the electron-dense rounded content of the CGs (arrowhead) is now extruded into the perivitelline space (PS) by exocytosis. Note the thick layer of the fertilization envelope (FE). (**B**) Treatment of immature oocytes with DTT for 70 min heavily alters the structure of the VL, to the extent that it prevents the visualization of the embedded microvilli (arrow). The internal structure of the CGs in these oocytes exhibits a more electron-dense content filamentous bodies (arrow). (**B′**) Five minutes after insemination, the elevated fertilization envelope (FE) shows partial disruption (arrow). The effect of DTT on the CGs is also evidenced by the altered morphology of their electron-dense content (arrowheads) exocytosed in the perivitelline space (PS). (**C**) The surface of an immature oocyte treated with 1-MA for 60 min and for an additional 10 min by DTT. Microvilli (arrow), cortical granules (CG). (**C′**) The short DTT treatment (10 min after stimulation with 1-MA for 60 min) was sufficient to alter the structure of the VL and CGs (arrowhead), as indicated by the rupture of the FE (arrows) in zygotes examined 5 min after insemination.

**Figure 7 biomolecules-13-01659-f007:**
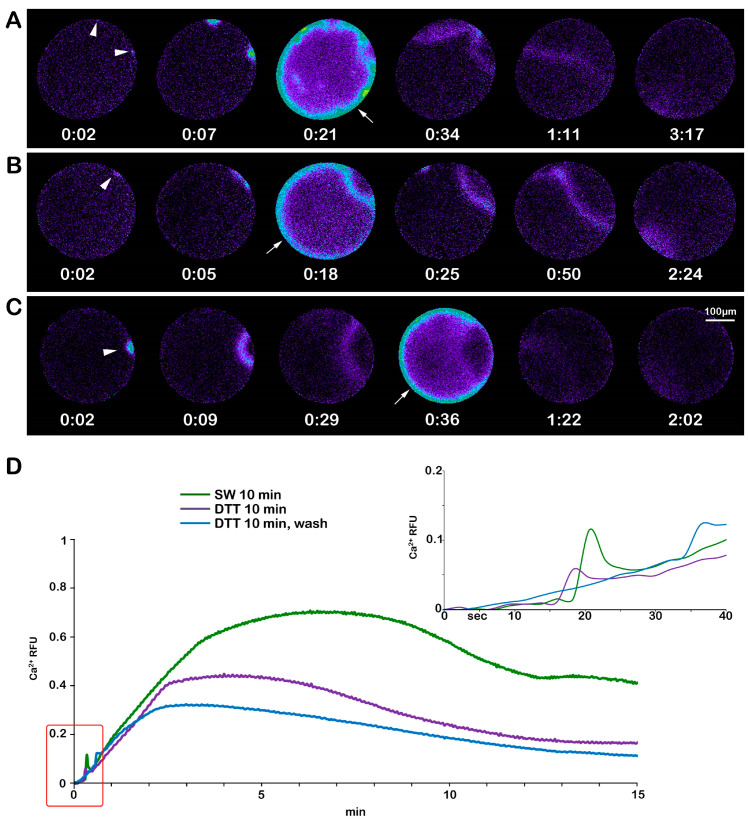
DTT preincubation alters the Ca^2+^ responses in starfish oocytes inseminated at the GV-stage. (**A**) The instantaneous increases in Ca^2+^ levels in an immature *A. aranciacus* oocyte after insemination in NSW. The Ca^2+^ signals detected upon sperm addition are the initiations of multiple Ca^2+^ waves (Ca^2+^ spots, arrowheads) due to stimulation by numerous sperm. Then, the cortical Ca^2+^ increase (the cortical flash, arrow) is triggered, occurring simultaneously at the periphery of the oocyte cortex. The multiple Ca^2+^ waves converge and run together to propagate to the opposite pole. (**B**) Ca^2+^ responses in an immature oocyte inseminated after 10 min of pretreatment with DTT. Ca^2+^ spot (arrowhead) and cortical flash (arrow) (**C**) Ca^2+^ responses in the same DTT-pretreated oocytes that were inseminated after washing in NSW. Ca^2+^ spot (arrowhead) and cortical flash (arrow). (**D**) Quantified Ca^2+^ trajectories represented in relative fluorescence unit (RFU) as defined in Section 2. The moment of the first detected Ca^2+^ signal was taken as t = 0, and the time is expressed in the minute and second format (mm: ss).

**Figure 8 biomolecules-13-01659-f008:**
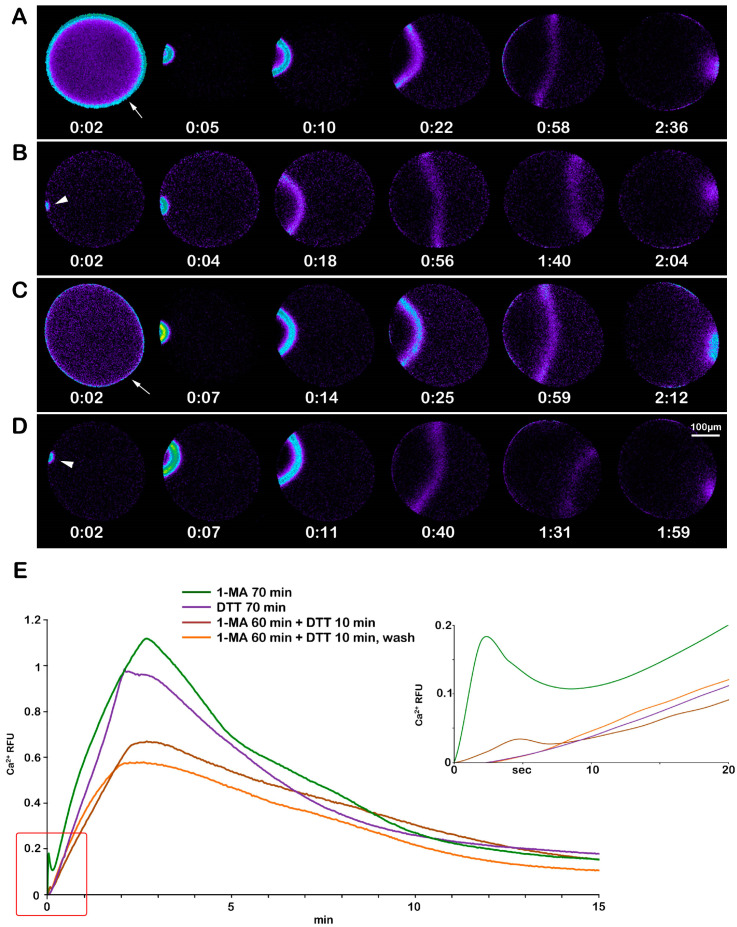
Alteration in the Ca^2+^ response at fertilization of immature oocytes induced to undergo maturation with DTT treatment. (**A**) The pseudocolor images of the Ca^2+^ response at fertilization of a control *A. aranciacus* GV-stage oocyte stimulated for 70 min with 1-MA. Sperm binding triggers the release of Ca^2+^ simultaneously at the periphery of the naturally matured egg (cortical flash, arrow), which is then followed by a Ca^2+^ wave (CW) that propagates from the sperm–egg fusion site to the opposite pole. (**B**) Treatment of immature oocytes with DTT for 70 min inhibits the sperm-induced CF and reduces the peak amplitude of the CW. The arrowhead indicates the initiation of the sperm-induced CW. (**C**) A short treatment with DTT (10 min) after stimulation of immature oocytes with 1-MA for 60 min (to induce GVBD) is sufficient to alter the Ca^2+^ response at fertilization. Cortical flash (arrow). (**D**) The washing out of DTT before insemination of eggs treated as in C still affects the Ca^2+^ response at fertilization. The arrowhead indicates the initiation of the sperm-induced CW. The pseudocolor fluorescent images and the graphs in (**E**) (inset) evidence the CF and CW being lower than the control in amplitude. (**E**) Quantified Ca^2+^ trajectories are represented in relative fluorescence unit (RFU).

**Table 1 biomolecules-13-01659-t001:** Effect of DTT on the Ca^2+^ response of immature oocytes upon insemination.

GV-Stage Oocytes (N = 3)	SW 10 min	DTT 10 min	DTT 10 min, Wash
Ca^2+^ response	16 out of 16	13 out of 16	11 out of 13
CF	12 out of 16	12 out of 16	10 out of 13
CF (RFU)	0.166 ± 0.010	0.047 ± 0.07	0.103 ± 0.04
Ca^2+^ wave (RFU)	0.59 ± 0.011	0.31 ± 0.09 *	0.26 ± 0.04 *
Traverse time (s)	134.04 ± 3.87	125.44 ± 7.1	132.68 ± 6.75
Time of activation (s)	19.6 ± 4.6	16.43 ± 2.4	19.9 ± 5.6
N spots	4.56 ± 0.87	3.07 ± 0.6 *	2.63 ± 0.75 *

Abbreviations: CF (cortical flash); RFU (relative fluorescence unit); * *p* < 0.01.

**Table 2 biomolecules-13-01659-t002:** Effect of DTT on the sperm-induced Ca^2+^ response of maturing oocytes.

Maturing Oocytes (N = 5)	1-MA 70 min	DTT 70 min	1-MA 60 min + DTT 10 min	1-MA 60 min + DTT 10 min, Wash
Ca^2+^ response	38 out of 38	9 out of 18	24 out of 25	17 out of 18
CF	32 out of 38	1 out of 18	18 out of 25	8 out of 18
CF (RFU)	0.095 ± 0.03	0.036	0.031 ± 0.01 *	0.063 ± 0.02 #
Ca^2+^ wave (RFU)	1.19 ± 0.13	0.89 ± 0.25 *	0.55 ± 0.19 *	0.70 ± 0.09 *
Traverse time (s)	129.9 ± 12.05	134.2 ± 16.23	117.7 ± 9.8	129.4 ± 5.6
Time of activation (s)	34.15 ± 8.9	203.45 ± 97.6 *	61.91 ± 35.03	35.62 ± 11.67

Abbreviations: CF (cortical flash); RFU (relative fluorescence unit); * *p* < 0.01; # *p* < 0.05.

## Data Availability

Data are contained within the article and Appendix A.

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
