# Peer review of "Dithiothreitol Affects the Fertilization Response in Immature and Maturing Starfish Oocytes"

_biomolecules, 2023, doi:10.3390/biom13111659_

Round 1

Reviewer 1 Report

Comments and Suggestions for Authors

1.       How does the reorganization of surface and cortical F-actin in immature starfish oocytes influence the normal oocyte maturation process leading to GVBD (germinal vesicle breakdown) and a fertilization response?

2.       What is the role of the maturation hormone 1-MA in triggering actin remodeling and Ca2+ changes in starfish oocytes, and how does this process resemble natural conditions?

3.       How does the structural organization of the extracellular vitelline layer covering the oocyte plasma membrane impact the fusion of multiple sperm with immature starfish oocytes?

4.       How does F-actin-based reconstitution of the oocyte cortex during the maturation of starfish oocytes contribute to appropriate electrical and Ca2+ responses upon sperm stimulation?

5.       What are the molecular mechanisms by which 1-MA resumes the meiotic cycle in starfish oocytes, and how does it compare to the artificial stimulation by DTT in terms of sulfhydryl contents in cortical proteins?

6.       What is the impact of DTT treatment on immature and maturing starfish oocytes' fertilization response, considering its effect on cortical actin and actin polymerization?

7.       How does the drastic depolymerization of cortical F-actin induced by DTT affect the subsequent reorganization and polymerization necessary for fertilization response in starfish oocytes, and how does it compare to other chemical agents like ionomycin and latrunculin-A?

Comments on the Quality of English Language

The sentences are generally well-structured, contributing to a coherent flow of ideas. However, there are a few instances where sentence structure could be improved for better clarity. Additionally, some sentences are quite lengthy, and breaking them down could enhance readability. The author effectively employs domain-specific terminology, showcasing a sound understanding of the scientific subject matter. Overall, the text is academically rigorous and communicates complex concepts effectively, but attention to sentence structure and length could further enhance its readability.

Author Response

Response to the Reviewers’ Comments

Reviewer #1:

  1. How does the reorganization of surface and cortical F-actin in immature starfish oocytes influence the normal oocyte maturation process leading to GVBD (germinal vesicle breakdown) and a fertilization response?

Our previous study with starfish oocytes microinjected with the antibody against actin-binding protein depactin demonstrated that the interference of the normal actin dynamics significantly delays the progression of meiotic maturation, as judged by completion of GVBD (Chun et al., 2013, BBRC, 441:301-7 doi: 10.1016/j.bbrc.2013.09.103). Prior alteration of the egg’s actin cytoskeleton influences many aspects of fertilization, as was described quite in detail in the Introduction (lines 45-51 of the original manuscript).

  1. What is the role of the maturation hormone 1-MA in triggering actin remodeling and Ca2+changes in starfish oocytes, and how does this process resemble natural conditions?

The mechanism by which 1-MA triggers F-actin remodeling and Ca2+ increase is largely unknown primarily because its receptor on the cell surface has not been identified yet. Nonetheless, it was demonstrated that heterotrimeric G-protein can recapitulate most effect of 1-MA (Sadler and Ruderman, 1998, Dev. Biol. 197:25-38. doi: 10.1006/dbio.1998.8869; Kyozuka et al. 2008, Dev. Biol. 320:426-35. doi: 10.1016/j.ydbio.2008.05.549). Whatever may be the mechanism, our original findings have shown that the purpose of the 1-MA in triggering actin remodeling, Ca2+ changes, and GVBD in starfish oocytes is to make mature eggs ready for successful fertilization. In nature (in the animal), 1-MA starts the maturation process, and the maturing oocytes are released and fertilized at sea between GVBD and the extrusion of the first polar body. In the laboratory, it has been possible for decades to follow the morphological and biochemical changes during the 1-MA-induced process. Our investigation has been focused on the role of the F-actin remodeling induced by 1-MA in regulating the fertilization process. We have slightly modified lines 591-596 of the Discussion.

  1. How does the structural organization of the extracellular vitelline layer covering the oocyte plasma membrane impact the fusion of multiple sperm with immature starfish oocytes?

Our previous publication (Santella et al. Zool. Lett. 2020) has shown initially that in immature starfish oocytes, the vitelline layer presents several holes through which multiple sperm can reach and fuse with the oocyte plasma membrane. The openings in the vitelline layer are reduced or disappear after the hormonal stimulation. The structural change of the vitelline layer, together with the cortical F-actin remodeling, prevents the attachment and fusion of supernumerary sperm. It ensures a normal fertilization response and monospermic penetration of the mature eggs. We had already explained in the original text of the manuscript this part:

Lines 55-57 from the Introduction: … Indeed, the tip of the F-actin-based acrosomal process of the fertilizing sperm triggered by components of the jelly coat (JC) can penetrate the vitelline layer (VL) endowed with openings and fuse with the cell plasma membrane [14,15,17,22-24].   

Lines 596-599 from the Discussion: …in immature (GV-stage) starfish oocytes, the fusion of multiple sperm with the oocyte plasma membrane [25,27,28] is firmly dependent on the structural organization of the extracellular vitelline layer covering the oocyte plasma membrane [14,15,17,21]…

  1.  How does F-actin-based reconstitution of the oocyte cortex during the maturation of starfish oocytes contribute to appropriate electrical and Ca2+responses upon sperm stimulation?

The question raised by the Reviewer#1 can be answered by referring to the papers that have been quoted in the original version of the manuscript:

Lines 599-614 from the Discussion: … the Ca2+ and electrophysiological changes after insemination of immature or maturing oocytes are linked to the length of microvilli (containing actin filaments) and to the structure of the CG positioned beneath the oocyte plasma membrane [8,10,62,63]. During the maturation of starfish oocytes, the F-actin-based reconstitution of the oocyte cortex (microvilli shortening and CG orientation) with the contribution of the nuclear components following GVBD establish the morpho-functional changes necessary to elicit appropriate electrical and Ca2+ response upon sperm stimulation [5,11-13,17,26]. In line with this, the Ca2+ signals during the 1-MA triggered cortical actin restructuration of starfish oocytes [9,13,64] are associated with cycles of decrease and increase in the rigidity (stiffness) of the maturing oocytes [37,65-68]. The rearrangement of the microvillar actin filaments that parallel the changes in potassium conductance in the regulation of the fertilization potential after the maturation process of starfish oocytes [62] could provide means for the changes in the electrical properties of the plasma membrane of maturing oocytes and for controlling the opening of channels that are located on the tip of microvilli and the influx pathway for external Ca2+ into the cell [69-71].

  1. What are the molecular mechanisms by which 1-MA resumes the meiotic cycle in starfish oocytes, and how does it compare to the artificial stimulation by DTT in terms of sulfhydryl contents in cortical proteins?

The molecular mechanisms by which 1-MA resumes the meiotic cycle in starfish oocytes have been only partially elucidated in 70-80s. The related reports were cited in the Introduction of the present contribution [Ref.1-2].

Lines 627-632 from the Discussion: To elucidate the molecular mechanism by which 1-MA resumes the meiotic cycle in starfish oocytes, previous studies compared the increase of the sulfhydryl (SH) contents in the cortical proteins of the oocytes after the incubation with 1-MA or DTT. The results showed that both 1-MA and DTT treatment caused an increase in the amount of pro-tein-SH in the oocytes, suggesting that a disulfide-reducing action of 1-MA is essential for resuming the maturation of A. pectinifera oocytes [35,36].

  1. What is the impact of DTT treatment on immature and maturing starfish oocytes' fertilization response, considering its effect on cortical actin and actin polymerization?

Lines 632-642 from the Discussion: In this contribution, we studied the in vivo effect of DTT on immature and maturing A. aranciacus oocytes by looking at their fertilization response since it is well known in vitro that the actin cytoskeletal changes, i.e., the depolymerization and polymerization states (whether G or F-actin) are influenced by the SH groups on cysteine whose oxidation helps to form an intermolecular disulfide (SS) bond to yield an actin dimer [74,75]. The results in Figs. 1 and 3 have shown that this was indeed the case. When immature A. aranciacus oocytes known to be polyspermic were treated for 10 minutes with DTT, the rate of sperm entry was reduced as a result of the alteration of the cortical actin of the oocyte induced by the disulfide-reducing agent. DTT treatment also affected the actin polymerization to form structures incorporating sperm in the cytoplasm (Fig. 3).

  1. How does the drastic depolymerization of cortical F-actin induced by DTT affect the subsequent reorganization and polymerization necessary for fertilization response in starfish oocytes, and how does it compare to other chemical agents like ionomycin and latrunculin-A?

Lines 663-684 from the Discussion: As a membrane-permeant reducing agent, DTT is expected to disrupt disulfide bonds in the proteins and other macromolecules on the surface and in the cortex of the oocytes. The ultrastructural TEM analyses have confirmed the action of DTT leading to partial destruction of the vitelline layer (VL) and microvilli morphology changes in immature and maturing A. aranciacus oocytes (Figs. 2 B and Fig. 6 B and C), as initially shown in A. pectinifera oocytes [36]. The striking alteration of microvillar morphology (Fig. 2 B, inset) may explain the inability of the F-actin-based long acrosomal process of the sperm to fuse with the oocyte plasma membrane [14,15] after reaching the preferential sites where microvilli are much longer due to their changed morphology (Fig. 2 B, inset). The structural alteration of the A. aranciacus oocyte surface by DTT also includes changes in the internal structure of the cortical granules (CG), as evidenced by the enhancement of the electron density of the material composing the core of the CG as well as the visualization of filament-like structures in the remaining part of the CG after the treatment with DTT (Figs. 6 B and D). The effect of DTT on the morpho-functionality of CGs is also expressed in the rosette-like and less compact electron-dense materials being extruded into the perivitelline space (PS) as posed to the round form that is routinuously found in control (arrowheads in Figs. 2 C and D and Figs. 6 A', B', C'). The changes in microvilli morphology and the structure of the CG are linked to the alteration of sperm-induced Ca2+ response in immature oocytes and oocytes stimulated with DTT treatments (Figs. 7 and 8). These findings add weight to our previous results in fertilized sea urchin eggs, demonstrating that microvillar actin filaments and CG contribute to shaping the Ca2+ response following sperm stimulation [73,79-82].

Lines 703-716 from the Discussion: As to the crucial role of the F-actin in controlling Ca2+ signaling in starfish oocytes, it may be worth mentioning the dramatic changes leading to a disintegration of the cortical actin, which is not followed by a rapid surface reconstitution when ionomycin and latrunculin-A (LAT-A) are used [26,61,84]. Previous results have shown that the intracellular Ca2+ increase after brief exposure of A. aranciacus immature oocytes to the Ca2+ ionophore ionomycin is accompanied by rapid and irreversible ultrastructural changes in the cortex of the oocyte, which include microvilli retraction and fusion of the cortical granules with other vesicles. That ionomycin pretreatment of the immature oocytes disrupted the functionality of the cortical actin cytoskeleton was evidenced by the com-promised Ca2+ release, CG exocytosis, and development after their physiological maturation with 1-MA. Similarly, artificial activation of mature A. aranciacus eggs with LAT-A, in addition to promoting an altered Ca2+ response (oscillations and not a single Ca2+ transient as experienced by the fertilized eggs), inhibit the subsequent microvilli elongation following the cortical actin polymerization [61,84,85].

Comments on the Quality of English Language

The sentences are generally well-structured, contributing to a coherent flow of ideas. However, there are a few instances where sentence structure could be improved for better clarity. Additionally, some sentences are quite lengthy, and breaking them down could enhance readability. The author effectively employs domain-specific terminology, showcasing a sound understanding of the scientific subject matter. Overall, the text is academically rigorous and communicates complex concepts effectively, but attention to sentence structure and length could further enhance its readability.

Some typos and redundant phrases have been indicated by the reviewers and the editors during the editorial communication. We appreciate what was pointed out. We took this opportunity to root out grammatical errors through the entire manuscript, and tried to make the sentences as brief as possible.

Reviewer 2 Report

Comments and Suggestions for Authors

This is a good-quality morphological study in which the authors showed that treatment of starfish oocytes with DTT—a disulfide-reducing agent known to induce oocyte maturation in various echinoderms—exhibits striking differences in the effects of DTT-induced and physiological/hormonal stimulation of meiotic resumption. The authors studied the in vivo effect of DTT on starfish (Astropecten aranciacus) oocytes, focusing on their capacity to mature and fertilize. They demonstrated that short-term DTT treatment reduced polyspermy and altered the sperm-induced Ca2+ response of the oocyte. Besides, the germinal vesicle breakdown (GVBD), indicating the end of meiotic prophase, does not occur even after 70 min of exposure to DTT. In vivo labeling experiments clearly demonstrated drastic depolymerization of the cortical F-actin in DTT-treated starfish oocytes at the GV stage, which results in impaired fertilization and embryo development. The unconventional use of vital staining for F-actin has been very useful and quite impressive. The effects of DTT were also confirmed at the ultrastructural level, and I have to emphasize the high quality of the electron microscopic images. It is important to note that some significant features were detected only at the electron microscopic level but were not visible under the light microscope, highlighting the importance of using electron microscopy in studying the functional morphology of cells.

I don’t have any major comments on the article, but I would like to draw the authors’ attention to some minor points.

1. I suggest that the authors edit the Conclusions section to make it more precise based on the findings, discussion, and further perspective. Some generalities—such as “Starfish oocytes isolated from the ovary are suitable experimental models...” (lines 724-725), “Oocyte maturation and fertilization are fundamental processes for animal development” (lines 740-741) and so on—would probably be better suited for the Introduction section. The Conclusion section should summarize briefly the most important findings of the study, with particular attention to pointing out prospects for future research.

2. What is the light ring-shaped area around the GV (Fig. 1B), which appears after 10-min DTT treatment? The same (?) is indicated by an arrow in Fig. 3 (B), but the legend does not explain what this arrow points to.

3. “The localized separation of the fertilization envelope” shown by an arrow in Fig. 1A (legend to the figure, line 236) is not evident in the EM image (Fig. 2A). I'm afraid this separation might be an artifact.

4. The CG label for cortical granules is missing in the Fig. 4, while “normal exocytosis of the cortical granules (CG)” is noted in the legend (lines 373-374). I doubt that exocytosis—normal or abnormal—can be observed in such images.

5. No “blue circle” indicated in the legend (line 431) and in the text (line 408) is visible in Fig. 5. The fertilization envelope (FE) also mentioned in this figure legend is not marked in the figure.

6. In the legend to Fig. 5, the explanation of the piece (C’) is missing.

7. Instead of Figure 8 in the caption to this figure it is again written Figure 7 (line 578).

8. The values on the vertical axes in the graphs (Figures 7D and 8E) are not deciphered.

9. When first mentioned, the name of the organism must be written in full (Astropecten aranciacus), while and the genus name should be abbreviated only later (A. aranciacus) — cf., lines 20 and 21; the name of the genus/species (Astropecten aranciacus) are not always italicized (e.g., lines 111, 184-185).

10. A few typos need to be corrected. Must be: a subscript for 6 in K3Fe(CN)6 (line 135); first capital letter for Ca2+ (line 581); pH instead of pHi (line 652).

11. This may go without saying to those in the field, but it seems it would be better to indicate how the sperm concentration was calculated (line 119).

Author Response

Reviewer 2:

This is a good-quality morphological study in which the authors showed that treatment of starfish oocytes with DTT—a disulfide-reducing agent known to induce oocyte maturation in various echinoderms—exhibits striking differences in the effects of DTT-induced and physiological/hormonal stimulation of meiotic resumption. The authors studied the in vivo effect of DTT on starfish (Astropecten aranciacus) oocytes, focusing on their capacity to mature and fertilize. They demonstrated that short-term DTT treatment reduced polyspermy and altered the sperm-induced Ca2+ response of the oocyte. Besides, the germinal vesicle breakdown (GVBD), indicating the end of meiotic prophase, does not occur even after 70 min of exposure to DTT. In vivo labeling experiments clearly demonstrated drastic depolymerization of the cortical F-actin in DTT-treated starfish oocytes at the GV stage, which results in impaired fertilization and embryo development. The unconventional use of vital staining for F-actin has been very useful and quite impressive. The effects of DTT were also confirmed at the ultrastructural level, and I have to emphasize the high quality of the electron microscopic images. It is important to note that some significant features were detected only at the electron microscopic level but were not visible under the light microscope, highlighting the importance of using electron microscopy in studying the functional morphology of cells.

We were pleased to read that the Reviewer acknowledged the good quality of the morpho-functional studies of the effect of DTT on the structural dynamic of the F-actin influencing the fertilization response of GV-stage immature oocytes and maturing oocytes. We also appreciated the comments on the importance of using electron microscopy to detect DTT-induced structural changes in the cells that can be missed out under a light microscope due to its limited resolution. 

I don’t have any major comments on the article, but I would like to draw the authors’ attention to some minor points.

  1. I suggest that the authors edit the Conclusions section to make it more precise based on the findings, discussion, and further perspective. Some generalities—such as “Starfish oocytes isolated from the ovary are suitable experimental models...” (lines 724-725), “Oocyte maturation and fertilization are fundamental processes for animal development” (lines 740-741) and so on—would probably be better suited for the Introduction section. The Conclusion section should summarize briefly the most important findings of the study, with particular attention to pointing out prospects for future research.

We modified the Conclusions section by eliminating the sentences highlighted by the Reviewer that were unsuitable for this section.

  1. What is the light ring-shaped area around the GV (Fig. 1B), which appears after 10-min DTT treatment? The same (?) is indicated by an arrow in Fig. 3 (B), but the legend does not explain what this arrow points to.

We thank the Reviewer for allowing us to clarify that the light ring-shaped area around the GV in Fig. 1 B shows the GV, which is out of focus. The image at the light microscope was made to show the fertilization cones induced by the entry of 2 spermatozoa in the immature starfish oocytes. By contrast, Fig. 3 B shows a confocal plane of a living A. aranciacus oocyte microinjected with Alexa-Phall568 to visualize actin filaments with and without treatment of the oocytes with DTT. In DTT-treated oocytes (5 out of 7), we observed that the actin filaments surrounding the GV were partially detached from the GV nuclear envelope (indicated by the arrow). We mentioned the results in the text (lines 290-291) but not in the legend of Fig. 3, so we have added it as suggested by the Reviewer.

  1. “The localized separation of the fertilization envelope” shown by an arrow in Fig. 1A (legend to the figure, line 236) is not evident in the EM image (Fig. 2A). I'm afraid this separation might be an artifact.

“The localized separation of the fertilization envelope,” shown by an arrow in Fig. 1A, is not present in the EM image in Fig. 2A because the image shows the surface of an unfertilized oocyte. At the same time, it is present in Fig. 2C (see FE). We confirm that our EM analysis descriptions are correct.

  1. The CG label for cortical granules is missing in the Fig. 4, while “normal exocytosis of the cortical granules (CG)” is noted in the legend (lines 373-374). I doubt that exocytosis—normal or abnormal—can be observed in such images.

We agree with the Reviewer that it is impossible to visualize the cortical granules exocytosis in the presented image, and we do not mean to do that. We wrote the sentence, “normal exocytosis of the cortical granules (CG)” because of the full elevation of the fertilization envelope presumably deriving from a normal exocytotic process. For clarity, we have eliminated the sentence in the legend of Fig. 4.

  1. No “blue circle” indicated in the legend (line 431) and in the text (line 408) is visible in Fig. 5. The fertilization envelope (FE) also mentioned in this figure legend is not marked in the figure.

The blue circle mentioned in the text and in the legend of Fig. 5 (A’ and A’’) shows the labelled DNA of the sperm head in the cytoplasm, indicated by a white arrow, and the online version of the manuscript will grant its visualization if a higher magnification is needed. However, since the term "circle" used to indicate the sperm inside the egg was not apparent to the Reviewer, we substituted "circle" with "spot" in the revised version of the manuscript.

  1. In the legend to Fig. 5, the explanation of the piece (C’) is missing.

We thank the Reviewer for picking up the missing explanation of panel C’. We have added it to the revised version of the manuscript.

  1. Instead of Figure 8 in the caption to this figure it is again written Figure 7 (line 578).

We have amended the typos and substituted 7 for 8 to indicate the correct figure number.

  1. The values on the vertical axes in the graphs (Figures 7D and 8E)are not deciphered.

We thank the Reviewer for highlighting the missing values on the vertical axes for both figures. We have inserted it in the revised version of the manuscript.

  1. When first mentioned, the name of the organism must be written in full(Astropecten aranciacus), while and the genus name should be abbreviated only later (A. aranciacus) — cf., lines 20 and 21; the name of the genus/species (Astropecten aranciacus) are not always italicized (e.g., lines 111, 184-185).

We are aware that the name of organisms, must be written in full when mentioned for the first time, and abbreviated only later. We thank the Reviewer for highlighting our mistake. We have revised it accordingly.

  1. A few typos need to be corrected. Must be: a subscript for 6 in K3Fe(CN)(line 135); first capital letter for Ca2+(line 581); pH instead of pHi (line 652).

We corrected the typos highlighted by the reviewer. We meant to write  “…..reduced pHi delays GVBD….. to indicate intracellular pH. We have substitute pHi with pHi.

  1. This may go without saying to those in the field, but it seems it would be better to indicate how the sperm concentration was calculated (line 119).

We thank the Reviewer for asking to indicate how the sperm concentration was calculated. We agree with the Reviewer that such information can interest readers who do not perform in vitro fertilization experiments. We have added the below procedure in the Materials and Methods of the revised version of the manuscript.

The sperm density was calculated according to the following procedure: A piece of male gonad was surgically removed from the animal and kept in a 1.5 ml Eppendorf tube. 3.5 μl of the dry sperm were added to 1 ml of natural seawater (NSW) in an Eppendorf tube (Diluted Sperm Stock). To determine the sperm density in the Diluted Sperm Stock, 2 μl aliquot was placed on the central groove of a hematocytometer, and then the number of sperm in a square millimeter were counted. The number obtained was 2,602 (=N). This number was multiplied by 10 in order to obtained the sperm count in 1 mm3 and then for 1000 to get the sperm count in 1 ml (Diluted Sperm Stock). Thus, the density of A. aranciacus dry sperm would be around 0.7x1010 per ml. For fertilization experiments, 2 µl dry sperm were dissolved in 1 ml NSW (500x dilution). From this solution, 10µl were added in 1 ml NSW containing the eggs (100x dilution). The final sperm concentration in the fertilization chamber would be 0.7x1010 per ml / 50,000 (dilution ratio) = ≈ 1x106 spermatozoa per ml.